# Effect of Preoperative Inflammatory Diet on Clinical and Oncologic Outcomes Following Colorectal Cancer Surgery

**DOI:** 10.3390/nu17091522

**Published:** 2025-04-30

**Authors:** Minjoon Kim, Haewon Kim, Kyeongeui Kim, Jaemin Cho, Woonkyung Jeong, Seongkyu Baek, Jaeho Lee, Sunguk Bae

**Affiliations:** 1Department of Medicine, Keimyung University School of Medicine, Daegu 42601, Republic of Korea; minjoon25@naver.com; 2Department of Nuclear Medicine, Keimyung University School of Medicine, Dongsan Medical Center, Daegu 42601, Republic of Korea; hwkim@dsmc.or.kr; 3Department of Surgery, Keimyung University School of Medicine, Dongsan Medical Center, Daegu 42601, Republic of Korea; ruddml205@gmail.com (K.K.); chojaemin09@naver.com (J.C.); shinycloud@dsmc.or.kr (W.J.); sgbbeak@dsmc.or.kr (S.B.); 4Department of Anatomy, Keimyung University School of Medicine, Dongsan Medical Center, Daegu 42601, Republic of Korea; anato82@dsmc.or.kr

**Keywords:** colorectal cancer, dietary inflammatory index, inflammatory diet, clinical outcomes, oncologic outcomes

## Abstract

**Objectives:** The dietary inflammatory index (DII), a validated tool for assessing the inflammatory potential of diet, has been widely identified as a significant risk factor for colorectal cancer (CRC). However, its role as a prognostic factor for CRC remains unexplored. This study examined the impact of preoperative dietary inflammation on clinical and oncologic outcomes following CRC surgery. **Methods**: The study population consisted of 126 patients who had surgical procedures for CRC and completed a food frequency questionnaire (FFQ) preoperatively between January 2018 and June 2020. **Results**: An optimal DII cut-off value of 0.90182 was used to categorize patients into the high-DII (*n* = 28) and low-DII (*n* = 98) groups. The high-DII group exhibited an older age (71.5 vs. 67.0, *p* = 0.020) and a significantly higher complication risk within 30 days postoperatively than the low-DII group (57.1% vs. 35.7%, *p* = 0.042). Other perioperative clinical outcomes did not demonstrate any significant differences between the two groups. The 5-year overall survival (OS) rates were 90.4% and 41.3% in the low-DII and high-DII groups, respectively, in univariate survival analysis (*p* = 0.044). However, no statistical difference was observed in the disease-free survival (DFS) rate. In the multivariate survival analysis, low-DII (hazard ratio [HR]: 0.118; 95% confidence interval [CI]: 0.023–0.613, *p* = 0.011) and M1 stage (HR: 10.910; 95% CI: 1.491–79.847, *p* = 0.019) were identified as independent prognostic factors for OS, while perineural invasion (HR: 3.495; 95% CI: 1.059–11.533, *p* = 0.040) served as an independent prognostic factor for DFS. **Conclusions:** A high preoperative DII score, indicative of an inflammatory dietary pattern, was correlated with increased postoperative complications and functioned as an independent prognostic indicator for OS.

## 1. Introduction

Globally, colorectal cancer (CRC) is the third-most common type of cancer and the second-leading cause of cancer-related mortality [1]. The incidence of CRC is influenced by both genetic and environmental factors, with well-known contributors such as dietary patterns, obesity, physical inactivity, smoking, and alcohol consumption [2]. Furthermore, the effects of these factors on CRC prognosis are currently being investigated [3,4,5,6].

In recent years, there has been a growing focus on individual dietary patterns as a modifiable factor that significantly influences cancer. Dietary patterns have been shown to affect cancer incidence in the general population [7,8,9,10] and influence cancer outcomes, including recurrence, survival, and life quality in cancer patients [11,12,13,14,15]. Because of the diversity and complexity of diets, various concepts of dietary patterns, such as vegetarian, high-meat, Western, Mediterranean, and inflammatory diets, have been used in different studies [7,8,9,10,11,12,13,14,15]. Among these, the inflammatory diet has been extensively studied due to its significant impact on chronic low-grade inflammation, which is known to contribute to the initiation and progression of several non-communicable diseases, including cancer, diabetes, and cardiovascular diseases [16,17]. The relationship between dietary components (e.g., macronutrients, micronutrients, and specific food items) and chronic systemic low-grade inflammation is well established [18].

The dietary inflammatory index (DII), derived from previous literature and population data, is used to estimate the pro- or anti-inflammatory potential of an individual’s diet [19]. The DII reflects the inflammatory potential of 45 food components, based on the previous literature investigating the relationships between various dietary constituents and six inflammatory cytokines (IL-1β, IL-4, IL-6, IL-10, TNF-α, and CRP). Designed to incorporate evidence from diverse human populations using various dietary assessment methods and study designs, the DII is not limited to specific dietary recommendations or regional foodways [20] and is also independent of population-specific characteristics. The DII has been employed in various studies addressing diseases where inflammation plays a key role, including obesity, type 2 diabetes, and cardiovascular disease [21,22,23]. In the field of oncology, it has been extensively studied as a potential risk factor for various malignancies, such as pancreatic, breast, and colorectal cancers [24,25,26,27].

Due to the pivotal role of prognostic factors in guiding clinical decisions and improving patient outcomes, continued research in this area is both necessary and timely. While prior studies have predominantly focused on clinical and pathological factors, there has been a progressive expansion of studies investigating the role of nutritional factors, such as sarcopenia, visceral obesity, and dietary patterns, in prognostic evaluation. Among these, the DII has emerged as a potential prognostic indicator in CRC. However, comprehensive studies that integrate DII with the full spectrum of clinicopathologic variables remain limited [28,29,30,31,32]. Therefore, we aimed to determine the prognostic significance of DII by performing a multivariate analysis on prospectively collected clinicopathologic and nutritional data. The results of this study may provide a foundation for evidence-based dietary guidance in the management of patients with CRC.

## 2. Materials and Methods

### 2.1. Ethical Considerations

Ethical approval for this study was obtained from the Institutional Review Board of Keimyung University Dongsan Medical Center (No. DSMC 2024-05-064). The process of collecting and examining data was conducted while considering ethical factors, specifically safeguarding the patients’ privacy rights. Due to the retrospective design of the study, wherein data were collected from past records and no direct interaction with participants occurred, the necessity for informed consent was waived.

### 2.2. Study Population

Electronic medical records of CRC patients who received surgical treatment at Dongsan Medical Center (DSMC) from January 2018 to June 2020 were reviewed. Among them, patients were excluded if they met the following exclusion criteria: (1) patients who did not consent to or could not respond to the food frequency questionnaire (FFQ), (2) patients with energy intake below 500 kcal or exceeding 10,000 kcal. As a result, this study enrolled 126 patients. The flow chart of patient selection is presented in Figure 1.

### 2.3. Data Collection and Definition

Prospectively compiled database and electronic medical records served as sources for the collection of variable data. The following baseline demographic information was documented just before surgery: age; sex; medical history of hypertension, diabetes mellitus, and abdominal surgery; carcinoembryonic antigen (CEA) and C-reactive protein (CRP) levels; height; weight; and tumor location. Furthermore, information on perioperative clinical outcomes and pathologic outcomes was documented postoperatively, including operation time; time taken to soft diet; length of hospital stay; combined resection of other organs; complications within 30 days from surgery; reoperation within 30 days from surgery; mortality within 30 days from surgery; administration of postoperative chemotherapy; types of complication; severity of complication assessed by Clavien–Dindo classification [33]; tumor, nodal, metastatic (TNM) stage of cancer; differentiation of cancer; number of retrieved lymph nodes; number of positive lymph nodes; tumor size; lymphovascular invasion status; perineural invasion status; and tumor-budding status.

When categorizing types of complications, acute kidney injury, pseudomembranous colitis, dysuria, respiratory complications, and acute pancreatitis were classified as medical complications, while ileus, anastomosis leakage, intra-abdominal abscess, wound infection, bleeding, and chyle leakage were classified as surgical complications. In cases where a patient exhibited two or more complications, the major complication was recorded. The eighth edition of the American Joint Committee on Cancer classification was used to assign TNM stages to all CRC patients. The time from surgery to either death from any cause or the last follow-up was used to define overall survival (OS). Disease-free survival (DFS) refers to the period from surgery to documented tumor recurrence, confirmed through radiological or histological evidence.

### 2.4. Dietary Inflammatory Index Calculation

The DII is a tool used to evaluate the inflammatory potential of individual dietary patterns [19]. The brief DII calculation process is as follows: (1) individual intake data obtained from the FFQ are standardized using Z-scores, calculated relative to a global reference database. The reference means and standard deviations for daily intake were developed by the original author using aggregated dietary data from 11 different countries. (2) Following conversion to percentile scores, Z scores are further transformed into centered percentile values, generating a symmetric distribution centered at zero and confined within the range of −1 to +1. (3) The centered percentile for each food parameter is multiplied by its corresponding inflammatory effect score, which was developed by the original author through an extensive literature review, to calculate the food parameter-specific DII score. (4) Summation of the DII scores for all individual food parameters yields the total DII score. A higher DII score reflects greater inflammatory potential of the diet, with scores above zero indicating a pro-inflammatory effect and scores below zero indicating an anti-inflammatory effect. The development process and calculation method of the DII are extensively covered elsewhere [19].

In this study, a 109-item semiquantitative FFQ was used to calculate the DII score (Appendix A) [34]. Patients reported the frequency of consumption and average intake amount for 109 types of foods over the past year. The FFQ also assessed whether there had been any changes in meal type or quantity during that period. If dietary changes were noted, participants were instructed to respond based on their dietary patterns from the year prior to the changes. The results of the FFQ were analyzed using CAN-Pro (Computer-aided nutritional analysis program) 4.0 software (The Korean Nutrition Society, Seoul, Republic of Korea), and data on the following 26 types of food parameters, which constitute the DII, could be obtained: energy, carbohydrate, total fat, protein, fiber, vitamin A, β-carotene, vitamin D, vitamin E, vitamin C, thiamine, riboflavin, niacin, vitamin B6, folic acid, vitamin B12, Mg, Fe, Zn, Se, cholesterol, saturated fat, monounsaturated fatty acids, polyunsaturated fatty acids, *n*-3 fatty acids, and *n*-6 fatty acids.

### 2.5. Statistical Analyses

The statistical analyses were conducted using SPSS (Statistical Package for the Social Sciences) 25.0 software (IBM Corp., Armonk, NY, USA), considering a *p*-value below 0.05 as indicative of statistical significance. To assess normality for variables, the Shapiro–Wilk test was conducted. Total numbers and percentages were used to summarize categorical variables, which were analyzed using the Chi-square test and Fisher’s exact test. Continuous variables were presented as mean ± standard deviation if they followed a normal distribution and analyzed using the independent *t*-test. Non-normally distributed variables were presented as median with interquartile range (IQR) and evaluated using the Mann–Whitney U test.

Patients were stratified into high-DII and low-DII groups according to their DII scores, using the Contal and O’Quigley method [35]. This statistical method was designed to find the optimal cut-off value of continuous variables by maximizing the log-rank statistics. It is known to be particularly useful in survival analysis with tied and censored data. In this study, the most optimal DII cut-off value for predicting OS was 0.90182.

To investigate the impact of each variable on patients’ OS and DFS, univariate survival analysis was performed using the Kaplan–Meier analysis. The number at risk represents the number of individuals who remain under follow-up at each time point and have not yet experienced the event of interest. Subsequently, variables with a *p*-value below 0.1 were collected for multivariate survival analysis: age, DII score, preoperative CEA, complications within 30 days of surgery, T stage, N stage, M stage, lymphovascular invasion, perineural invasion, and tumor budding. The multivariate survival analysis was conducted by Cox proportional hazards regression analysis to estimate multivariable-adjusted hazard ratios (HRs) and 95% confidence intervals (95% CIs). The proportional hazard (PH) assumption was tested by the Schoenfeld residual test. No evidence of PH violation was found by Schoenfeld residuals (all *p* > 0.05; global *p* = 0.10).

## 3. Results

### 3.1. Patient Characteristics

This study included a total of 126 patients. Based on the optimal DII cut-off value determined by the Contal and O’Quigley method, 28 patients (22.2%) were assigned to the high-DII group and 98 patients (77.8%) to the low-DII group. The high-DII group consists of individuals adhering to diets with higher inflammatory potential, while the low-DII group includes those following dietary patterns with anti-inflammatory potential. The baseline characteristics of each group are presented in Table 1. The median and interquartile range of the overall DII score for the high-DII group was 2.29 (1.35–2.88) and −2.06 (−3.33–−0.88) for the low-DII group. A significant difference in age was observed, where the median ages of the high-DII and low-DII groups were 71.50 (65.00–78.00) and 67.00 (57.00–74.00) years (*p* = 0.020), respectively. In contrast, no significant differences were noted between the two groups in the other variables.

### 3.2. Perioperative Clinical Outcomes

Perioperative clinical outcomes are presented in Table 2. The number of complications occurring within 30 days of surgery was statistically different between the groups, with 16 cases (57.1%) in the high-DII group and 35 cases (35.7%) in the low-DII group (*p* = 0.042). Meanwhile, no significant differences were found in other factors.

The details of complications occurring within 30 days from surgery are presented in Table 3. Regarding the types of complications, bleeding was the most common complication in the high-DII group at a rate of 14.3% (four cases). Conversely, in the low-DII group, ileus was the most common one, at a rate of 7.1% (seven cases). Among these, only bleeding showed a significant difference between the two groups (*p* = 0.043). The complications were further categorized into medical and surgical; no significant differences were noted between the groups in either category. Furthermore, no significant difference was observed in the incidence of severe complications, as indicated by a Clavien–Dindo classification of 3a or higher for both groups.

### 3.3. Postoperative Pathologic Outcomes

Postoperative pathologic outcomes are presented in Table 4. No significant differences were observed between the two groups in terms of TNM, differentiation, retrieved lymph nodes, positive lymph nodes, tumor size, lymphovascular invasion, perineural invasion, and tumor budding.

### 3.4. Oncologic Outcomes

The high-DII group had a median follow-up duration of 41 months, compared to 42 months in the low-DII group. The low-DII group showed significantly higher 5-year OS rates than the high-DII group (90.4% vs. 41.3%, *p* = 0.044), but the 5-year DFS rates did not differ significantly between the two groups (76.4% vs. 72.3%, *p* = 0.850). The Kaplan–Meier curves of OS and DFS for each DII group are depicted in Figure 2. A total of 26 patients had a recurrence during follow-up, and the pattern of recurrence did not significantly differ between the two groups. The high-DII group had six cases of recurrence (four cases of systemic and two of local recurrence), while the low-DII group had 20 cases of recurrence (16 cases of systemic and five of local recurrences; one patient had both types of recurrence).

### 3.5. Univariate and Multivariate Analyses of Prognostic Factors Associated with Oncologic Outcomes

The results of univariate survival analysis conducted using the Kaplan–Meier analysis are presented in Table 5. The prognostic factors associated with OS included DII score, preoperative CEA, complications within 30 days of surgery, M stage, lymphovascular invasion, and perineural invasion. Meanwhile, complications within 30 days of surgery, T stage, N stage, M stage, lymphovascular invasion, and perineural invasion were found to be associated with DFS.

The results of multivariate survival analysis conducted using Cox proportional hazards regression analysis are presented in Table 6. The low-DII group (HR: 0.118; 95% CI: 0.023–0.613, *p* = 0.011) and the M1 stage (HR: 10.910; 95% CI: 1.491–79.847, *p* = 0.019) were identified as independent prognostic factors for OS. Additionally, perineural invasion was identified as an independent prognostic factor for DFS (HR: 3.495; 95% CI: 1.059–11.533, *p* = 0.040).

## 4. Discussion

In this study, the group that consumed a more inflammatory diet showed significantly worse postoperative oncologic outcomes and greater postoperative complication rates compared to the group consuming a less inflammatory diet. The univariate survival analysis revealed that the DII was significantly associated with OS. Furthermore, multivariate survival analysis confirmed that the DII, along with M stage, was an independent prognostic factor for OS. To the best of our knowledge, this study is the first to analyze the association between the DII and postoperative complications from cancer surgery and one of the few studies to focus on oncologic outcomes following surgery in CRC patients.

The DII is reportedly associated with oncologic outcomes in other cancers, such as breast and primary liver cancers [36,37]. Regarding CRC, two previous studies have also reported a significant association between DII and oncologic outcomes [28,31]. Galas et al. found that the DII was associated with the 3-year survival of patients without distant metastases [28], while Zheng et al. reported a significant correlation between DII and all-cause mortality in postmenopausal women [31]. However, three other studies did not find a statistically significant association [29,30,32]. These conflicting findings warrant further investigation. However, these studies have several limitations compared to the current one. None of them included important prognostic factors such as TNM stage, tumor budding, lymphovascular invasion, and perineural invasion in the adjustment in the multivariate survival analysis [28,29,30,31,32]. Additionally, some studies used a previous version of the DII [28] or a modified version of the DII created by other researchers [32], making their direct comparison with the majority of other studies challenging. Moreover, some studies did not account for surgical treatment of CRC [29,31,32], and in some cases, only postmenopausal women were included in the study cohort [31]. Unlike previous studies, our research, based on the standard version of the DII, specifically targeted patients who had undergone surgical treatment and included an analysis of key prognostic factors. This approach strengthened the evidence for a relationship between the DII and oncologic outcomes in CRC, making it a critical contribution to the field.

Chronic inflammation has been demonstrated to have a substantial impact on postoperative complications in a variety of surgical settings, with a particular emphasis on cancer interventions [38]. Chronic inflammation is strongly influenced by the consumption of an inflammatory diet, which is typically high in processed foods, sugars, refined carbohydrates, and unhealthy lipids [18]. Inflammatory diets, defined by a high intake of processed meats, refined carbohydrates, and low consumption of fruits and vegetables, are associated with increased inflammation and may result in a higher number of postoperative complications [39]. A prior study that investigated the relationship between the DII and clinical outcomes in CRC patients reported a significant association between the DII and length of hospitalization [40]. The findings of the present study showed a notable difference in postoperative complication rates across the two groups, particularly with regard to bleeding. Despite the fact that the small sample size of the study makes it challenging to draw a definitive conclusion regarding the relationship and mechanism between an inflammatory diet and postoperative complications, chronic inflammation in the host as a result of an inflammatory diet could be considered to have an impact on the immediate clinical outcome after surgery. Further research is required in the future in this regard.

Preoperative CRP levels are generally known to be associated with survival in patients with CRC [6,41]. However, in this study, preoperative CRP levels did not show a statistically significant difference between the high-DII and low-DII groups. Additionally, in the univariate survival analysis, CRP did not demonstrate prognostic significance, similar to the DII. However, as the DII is based on its relationship with six inflammatory cytokines, the interpretation of CRP alone may be limited. Previous studies have also reported inconsistent findings, with some demonstrating a significant correlation between CRP and the DII, while others have found no meaningful association [42,43]. It is necessary to conduct further studies that incorporate all six inflammatory cytokines.

Recently, gut microbiota has emerged as a crucial link between diet and CRC [44,45,46,47,48]. Diet plays a key role in shaping the structure and diversity of the gut microbiota [49,50], and the same applies to an inflammatory diet [51,52]. Moreover, the gut microbiota maintains a close interaction with the immune system, and dysbiosis (alterations in gut microbiota composition) can initiate chronic inflammation, affecting cytokine-signaling pathways [44,47]. A study demonstrated that the pro-inflammatory potential of the intestine can be predicted by the DII score, which is correlated to microbiome diversity [53]. Therefore, diet–microbiome interactions may be utilized as potential targets for the reduction of chronic inflammation, a significant factor in the development of numerous chronic conditions, including obesity, inflammatory bowel disease, and cardiovascular diseases [54]. This expanding body of evidence indicates that dietary modifications have the potential to affect the gut microbiome, which could have implications for the management of inflammatory-related diseases and the reduction of inflammation. Nevertheless, additional research is required to gain a comprehensive understanding of the specific mechanisms and to establish precise dietary recommendations.

In addition to the inflammatory diet, various dietary patterns have been utilized in studies investigating their association with the oncologic outcomes of CRC. Consumption of a Western diet has been linked to adverse outcomes, including increased recurrence and mortality, in stage III colon cancer patients [11]. Similarly, dietary patterns with red and processed meat have shown significant associations with increased risks of CRC recurrence and overall mortality [12]. Additionally, strong adherence to the Mediterranean and Nordic diets has been linked to an improved OS in long-term CRC survivors [13]. Alongside these findings, as indicated by the results of this study, if dietary patterns at the time of diagnosis are identified as independent prognostic factors for CRC surgery, the need for healthy dietary patterns in the general population must be emphasized, necessitating a systematic preventive medical approach and further research. Furthermore, growing evidence highlights the potential impact of dietary modifications on cancer outcomes [55]. Building upon the significance of preoperative dietary patterns demonstrated in our study, future research is expected to strengthen the role of such interventions, potentially expanding the focus from preoperative dietary patterns to comprehensive nutritional approaches for optimizing oncologic outcomes.

This study has some limitations. First, this study was retrospectively conducted at a single institution, resulting in a limited study population. Subsequent multi-center prospective studies, including a larger study population, should be conducted. Second, the FFQ results were analyzed using the CAN-Pro 4.0 software, which can only calculate the quantities of predetermined food components. Therefore, of the 45 food parameters, only 26 were used in calculating the DII. Although this method may have reduced the predictive value of the DII, a previous study reported no significant difference in predictability between the DIIs calculated with 28 parameters and 44 parameters [56]. Furthermore, previous studies on CRC and DII used a wide range of food parameters [27], ranging from a minimum of 18 parameters [57]. Third, this study focused solely on the clinical and oncologic outcomes of CRC, without addressing the biological and molecular mechanisms underlying the impact of inflammatory diets. Future research should include molecular studies using tumor samples or animal models, as well as investigations at the microbiome level. Finally, because the FFQ assesses dietary patterns only over the past year and can detect dietary changes only if they occurred within that specific timeframe, it cannot reflect the inflammatory potential of dietary patterns prior to that period or after FFQ administration. Therefore, for optimal results, patients who have undergone significant changes in their dietary patterns before the recall period need to be excluded from the study, and follow-up FFQs must be conducted annually to confirm whether the dietary pattern has undergone any changes. Furthermore, as this study did not examine the role of postoperative dietary inflammation or dietary modifications, future research is needed to address these aspects.

## 5. Conclusions

A high preoperative DII was associated with a higher incidence of postoperative complications and worse oncologic outcomes. Furthermore, it remained an independent prognostic factor for overall survival in the multivariate analysis.

## Figures and Tables

**Figure 1 nutrients-17-01522-f001:**
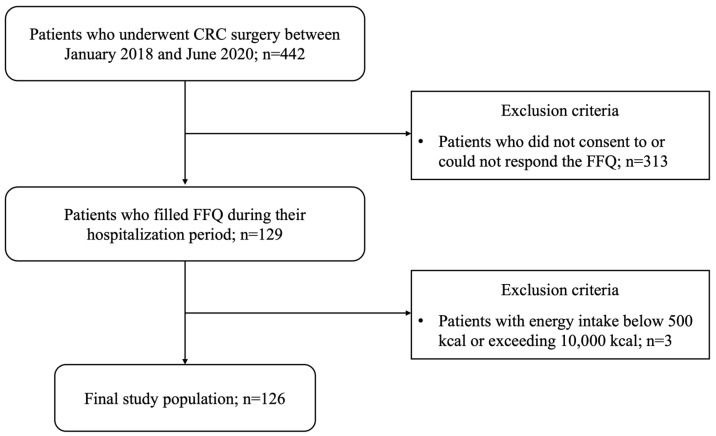
Study flow chart.

**Figure 2 nutrients-17-01522-f002:**
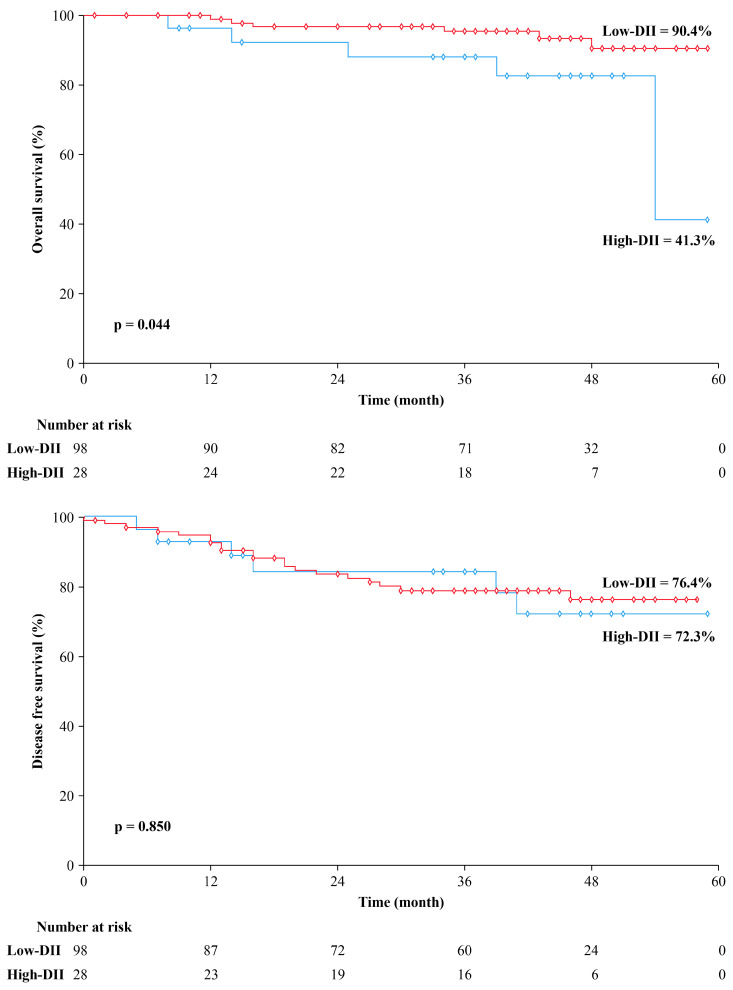
Kaplan–Meier curve for overall survival and disease-free survival stratified by dietary inflammatory index (DII).

**Table 1 nutrients-17-01522-t001:** Patient characteristics.

	High-DII Group (*n* = 28)	Low-DII Group (*n* = 98)	*p* Value
Overall DII score	2.29 (1.35–2.88)	−2.06 (−3.33–−0.88)	<0.001
Age (years)	71.50 (65.00–78.00)	67.00 (57.00–74.00)	0.020
Sex			0.442
Male	14 (50.0)	57 (58.2)	
Female	14 (50.0)	41 (41.8)	
Medical history			
Hypertension	16 (57.1)	42 (42.9)	0.181
Diabetes mellitus	7 (25.0)	18 (18.4)	0.438
Abdominal surgery	6 (21.4)	35 (35.7)	0.155
Preoperative CEA > 5 ng/mL	8 (28.6)	22 (22.7)	0.520
Preoperative CRP	0.20 (0.03–0.47)	0.11 (0.05–0.61)	0.776
BMI (kg/m^2^)	23.26 ± 3.2	24.15 ± 3.5	0.237
Location of tumor			0.841
Rt	10 (35.7)	33 (33.7)	
Lt	18 (64.3)	65 (66.3)	

Values are presented as mean ± standard deviation or median (IQR) or number (%). DII = dietary inflammatory index; CEA = carcinoembryonic antigen; CRP = C-reactive protein; BMI = body mass index.

**Table 2 nutrients-17-01522-t002:** Perioperative clinical outcomes.

	High-DII Group (*n* = 28)	Low-DII Group (*n* = 98)	*p* Value
Operation time (min)	227.00 (167.50–280.00)	190.00 (147.00–255.00)	0.057
Time taken to soft diet (days)	6.00 (4.00–7.50)	6.00 (4.00–8.00)	0.999
Length of hospital stay (days)	9.50 (7.00–10.50)	9.00 (8.00–10.00)	0.574
Combined resection of other organs	6 (21.4)	10 (10.2)	0.116
Complications within 30 days of surgery	16 (57.1)	35 (35.7)	0.042
Reoperation within 30 days of surgery	2 (7.1)	3 (3.1)	0.308
Mortality within 30 days of surgery	0 (0)	0 (0)	-
Postoperative chemotherapy	15 (53.6)	61 (62.2)	0.408

Values are presented as median (IQR) or number (%).

**Table 3 nutrients-17-01522-t003:** Details of complications occurring within 30 days of surgery.

	High-DII Group (*n* = 28)	Low-DII Group (*n* = 98)	*p* Value
Total number	16 (57.1)	35 (35.7)	0.042
Types			
Acute kidney injury	0 (0)	1 (1.0)	1.000
Ileus	2 (7.1)	7 (7.1)	1.000
Pseudomembranous colitis	0 (0)	2 (2.0)	1.000
Anastomosis leakage	1 (3.6)	4 (4.1)	1.000
Intra-abdominal abscess	1 (3.6)	1 (1.0)	0.396
Dysuria	3 (10.7)	5 (5.1)	0.375
Wound infection	2 (7.1)	5 (5.1)	0.651
Bleeding	4 (14.3)	3 (3.1)	0.043
Respiratory complications	1 (3.6)	0 (0)	0.222
Chyle leakage	2 (7.1)	6 (6.1)	1.000
Acute pancreatitis	0 (0)	1 (1.0)	1.000
Classification			
Medical Complications	4 (14.3)	9 (9.2)	0.483
Surgical Complications	12 (42.9)	26 (26.5)	0.097
Severity (Clavien-Dindo classification)			
CD1	6 (21.4)	13 (13.3)	0.287
CD2	7 (25.0)	18 (18.4)	0.438
CD3a	1 (3.6)	0 (0)	0.222
CD3b	1 (3.6)	2 (2.0)	0.533
CD4a	0 (0)	2 (2.0)	1.000
CD4b	1 (3.6)	0 (0)	0.222
CD ≥ 3a	3 (10.7)	4 (4.1)	0.183

Values are presented as number (%).

**Table 4 nutrients-17-01522-t004:** Postoperative pathologic outcomes.

	High-DII Group (*n* = 28)	Low-DII Group (*n* = 98)	*p* Value
Tumor stage			0.762
T0, T1, T2	10 (35.7)	32 (32.7)	
T3, T4	18 (64.3)	66 (67.3)	
Nodal stage			0.847
N0	16 (57.1)	58 (59.2)	
N1, N2	12 (42.9)	40 (40.8)	
Metastatic stage			1.000
M0	26 (92.9)	92 (93.9)	
M1	2 (7.1)	6 (6.1)	
Differentiation			0.227
Well, moderately differentiated	24 (85.7)	92 (93.9)	
Poorly, mucinous differentiated	4 (14.3)	6 (6.1)	
Retrieved LNs	16.50 (13.00–21.50)	18.00 (13.00–29.00)	0.544
Positive LNs	0.00 (0.00–2.50)	0.00 (0.00–1.00)	0.256
Tumor size (cm)	4.45 (2.85–6.45)	3.80 (2.50–4.70)	0.253
Lymphovascular invasion	10 (38.5)	37 (38.1)	0.976
Perineural invasion	9 (33.3)	25 (26.0)	0.454
Tumor budding	16 (59.3)	59 (62.8)	0.741

Values are presented as median (IQR) or number (%).

**Table 5 nutrients-17-01522-t005:** Prognostic factors of OS and DFS in univariate analysis.

Prognostic Factor	N	Overall Survival (5 Years, %)	*p* Value	Disease-Free Survival (5 Years, %)	*p* Value
Age (years)			0.097		0.302
≤65	52	64.8		69.8	
>65	74	90.8		79.2	
Sex			0.482		0.675
Male	71	83.2		75.3	
Female	55	75.3		75.7	
DII score			0.044		0.850
≤0.90182	98	90.4		76.4	
>0.90182	28	41.3		72.3	
BMI			0.548		0.676
≤25	82	78.1		77.4	
>25	44	85.2		72.0	
History of hypertension			0.168		0.296
Negative	68	82.5		73.1	
Positive	58	81.4		78.1	
History of diabetes mellitus			0.817		0.405
Negative	101	79.1		74.1	
Positive	25	92.0		77.0	
Tumor location			0.342		0.241
Rt side	43	84.6		67.7	
Lt side	83	80.5		78.8	
Preoperative CEA			0.017		0.103
≤5	95	84.9		79.7	
>5	30	71.8		65.1	
Preoperative CRP			0.563		0.748
≤0.3	75	77.8		72.0	
>0.3	37	94.3		79.8	
Complications within 30 days of surgery			0.011		0.008
Negative	75	87.5		82.7	
Positive	51	75.0		64.6	
Differentiation			0.113		0.467
Well, moderately differentiated	116	82.0		77.7	
Poorly, mucinous differentiated	10	78.8		48.0	
T stage			0.086		0.011
T0, T1, T2	42	97.4		90.7	
T3, T4	84	72.6		68.0	
N stage			0.260		0.008
N0	74	78.0		84.8	
N1, N2	52	84.9		61.1	
M stage			0.001		0.013
M0	118	84.4		77.3	
M1	8	35.7		46.9	
Lymphovascular invasion			0.003		0.004
Negative	76	86.5		85.1	
Positive	47	77.2		58.4	
Perineural invasion			0.001		0.001
Negative	89	86.8		84.5	
Positive	34	73.1		49.1	
Tumor budding			0.059		0.452
Negative	46	96.2		79.9	
Positive	75	70.5		71.2	

DII = dietary inflammatory index; BMI = body mass index; CEA = carcinoembryonic antigen; CRP = C-reactive protein.

**Table 6 nutrients-17-01522-t006:** Prognostic factors of OS and DFS in multivariate analysis.

Variables	Reference Category	Overall Survival		Disease-Free Survival	
		HR (95% CI)	*p* Value	HR (95% CI)	*p* Value
Age (years)		0.314 (0.060–1.658)	0.172	1.118 (0.434–2.885)	0.817
>65	≤65				
DII score		0.118 (0.023–0.613)	0.011	1.246 (0.441–3.520)	0.678
≤0.90182	>0.90182				
Preoperative CEA		0.982 (0.190–5.072)	0.982	1.258 (0.461–3.438)	0.654
≤5	>5				
Complications within 30 days from surgery		4.643 (0.548–39.363)	0.159	2.352 (0.957–5.777)	0.062
Positive	Negative				
T stage		104314.920 (0.000–5.100)	0.958	1.878 (0.478–7.376)	0.367
T3, T4	T0, T1, T2				
N stage		0.183 (0.023–1.424)	0.105	1.175 (0.426–3.243)	0.756
N1, N2	N0				
M stage		10.910 (1.491–79.847)	0.019	2.325 (0.634–8.525)	0.203
M1	M0				
Lymphovascular invasion		3.904 (0.431–35.366)	0.226	1.963 (0.727–5.299)	0.183
Positive	Negative				
Perineural invasion		8.360 (0.808–86.452)	0.075	3.495 (1.059–11.533)	0.040
Positive	Negative				
Tumor budding		3.818 (0.381–38.247)	0.254	0.558 (0.184–1.686)	0.301
Positive	Negative				

DII = dietary inflammatory index; CEA = carcinoembryonic antigen.

## Data Availability

Data supporting the findings of this study are available upon request from the corresponding authors. The analyzed data are not available in public because of privacy or ethical restrictions.

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
