# Peer review of "Effect of Preoperative Inflammatory Diet on Clinical and Oncologic Outcomes Following Colorectal Cancer Surgery"

_nutrients, 2025, doi:10.3390/nu17091522_

Round 1

Reviewer 1 Report

Comments and Suggestions for Authors

The authors make a contribution to the research literature in this field. I found the results very interesting. But there are a few questions and comments for your consideration.

  1. The gap and significance of this study should be clearly explained.
  2. Did the researchers compare baseline characteristics between study completers and non-respondents? Please add.
  3. Please add the scientific basis for the sample size of the research subjects.
  4. In the part of statistical analyses, the proportional hazards (PH) hypothesis is a core premise of Cox regression and need to be stated in the method.
  5. The clinical relevance of DII cut-off value should be supplemented. For example, a DII score above 0.90 may indicate a more pro-inflammatory dietary pattern…
  6. In lines 189, 201, and 208, please verify if Table 2-4 contain solely median (IQR) or number (%) values and align the textual statements accordingly.
  7. Please elaborate on the implications for future research and healthcare guidance in discussion section.
  8. Minor editing of English language required.

Author Response

  1. The gap and significance of this study should be clearly explained.
    : We appreciate the reviewer's feedback. Prior to our study, most existing research focused on the association between the DII and the incidence of colorectal cancer, while only a limited number of studies investigated clinical or oncologic outcomes. Furthermore, even among those few studies, important prognostic factors (such as TNM stage, tumor budding, lymphovascular invasion, and perineural invasion) were not included in the adjustment in multivariate survival analysis. Therefore, our study strengthens the existing evidence regarding the relationship between the DII and oncologic outcomes in colorectal cancer, making it a meaningful contribution to the field. In addition, to the best of our knowledge, this study is the first to analyze the association between the DII and postoperative complications from cancer surgery.
    This issue has been addressed in the Discussion section as follows. (line 257-259, line 274-278)
    “To the best of our knowledge, this study is the first to analyze the association between the DII and postoperative complications from cancer surgery and one of the few studies to focus on oncologic outcomes following surgery in CRC patients.”
    “Unlike previous studies, our research, based on the standard version of the DII, specifically targeted patients who had undergone surgical treatment and included an analysis of key prognostic factors. This approach strengthened the evidence for a relationship be-tween the DII and oncologic outcomes in CRC, making it a critical contribution to the field.”

  2. Did the researchers compare baseline characteristics between study completers and non-respondents? Please add.
    : We appreciate the reviewer's suggestions for improvement. We have added the comparison of baseline characteristics between study completers and non-respondents as a separate file. Please see the attachment.

  3. Please add the scientific basis for the sample size of the research subjects.
    : We sincerely appreciate your insightful comment. During the initial study design phase, we performed a sample size calculation using the ‘ssizeCT.default’ function in R. Based on referenced values, we set pC(probability of failure in the control group over the maximum time period of the study) at 0.37 [1] and the relative risk (RR) at 0.49 [2], which indicated that a total study population of 232 participants would be required. Anticipating a response rate of approximately 50% for the food frequency questionnaire (FFQ), we aimed to recruit 400–500 patients in total.
    Throughout the study period, we recommended the FFQ to all eligible patients. However, given the extensive number of items (109 questions) and the estimated completion time of around 50 minutes, many patients declined to participate, resulting in a lower-than-expected response rate despite our efforts.
    To address the limitation of the insufficient sample size, we employed the Contal and O'Quigley method, a statistical technique that identifies the optimal cut-off value to maximize the survival difference between the two groups.
    Reference
    [1] Siegel RL, Giaquinto AN, Jemal A. Cancer statistics, 2024. CA Cancer J Clin. 2024 Jan-Feb;74(1):12-49. doi: 10.3322/caac.21820. Epub 2024 Jan 17. Erratum in: CA Cancer J Clin. 2024 Mar-Apr;74(2):203. doi: 10.3322/caac.21830. PMID: 38230766.
    [2] Zheng J, Tabung FK, Zhang J, Murphy EA, Shivappa N, Ockene JK, Caan B, Kroenke CH, Hébert JR, Steck SE. Post-cancer diagnosis dietary inflammatory potential is associated with survival among women diagnosed with colorectal cancer in the Women's Health Initiative. Eur J Nutr. 2020 Apr;59(3):965-977. doi: 10.1007/s00394-019-01956-z. Epub 2019 Apr 6. PMID: 30955051; PMCID: PMC6778721.

  4. In the part of statistical analyses, the proportional hazards (PH) hypothesis is a core premise of Cox regression and need to be stated in the method.
    : We sincerely appreciate your insightful comment on this point. We have addressed this by verifying the issue using the ‘cox.zph’ function in R, and have added the following paragraph to the Materials and Methods section. (line 170-172)
    “The proportional hazard (PH) assumption was tested by the Schoenfeld residual test. No evidence of PH violation was found by Schoenfeld residuals (all p>0.05; global p=0.10).”

  5. The clinical relevance of DII cut-off value should be supplemented. For example, a DII score above 0.90 may indicate a more pro-inflammatory dietary pattern.
    : In accordance with your advice, we have explained the clinical relevance of the two groups divided by the DII cut-off value with the following statement in Results section. (line 177-179)
    “The high-DII group consists of individuals adhering to diets with higher inflammatory potential, while the low-DII group includes those following dietary patterns with anti-inflammatory potential.”

  6. In lines 189, 201, and 208, please verify if Table 2-4 contain solely median (IQR) or number (%) values and align the textual statements accordingly.
    : Thanks to your thoughtful comment, we have reviewed the contents of each table and removed unnecessary textual statements accordingly.

  7. Please elaborate on the implications for future research and healthcare guidance in discussion section.
    : Thank you for your valuable comments. To connect the findings of our study with suggestions for future research, we have added the following paragraph to the Discussion section. (line 321-326)
    “Furthermore, growing evidence highlights the potential impact of dietary modifications on cancer outcomes [52]. Building upon the significance of preoperative dietary patterns demonstrated in our study, future research is expected to strengthen the role of such interventions, potentially expanding the focus from preoperative dietary patterns to comprehensive nutritional approaches for optimizing oncologic outcomes.”

  8. Minor editing of English language required.
    : Thanks to your comment. This manuscript has undergone professional English editing before, and we have also carefully re-examined the sections that were modified after the editing process. If you could kindly point out any awkward or unclear parts, we will be glad to make the necessary corrections.

Reviewer 2 Report

Comments and Suggestions for Authors

The present study addresses a timely and relevant topic; however, several aspects require further clarification and refinement to strengthen the manuscript. The Introduction section could benefit from a more comprehensive overview of previous related studies, potentially including other cancer types, to better contextualize the current work. Additionally, the authors should clearly articulate the unique contribution of their study to the existing literature.

In the Materials and Methods section, the description of the DII calculation method lacks sufficient detail and does not fully adhere to the journal’s methodological reporting guidelines. For enhanced clarity, it is recommended that an example of the food frequency questionnaire (FFQ) used in the study be included in the Supplementary Materials.

Furthermore, it would have been valuable to assess preoperative levels of pro- and anti-inflammatory cytokines, as these are referenced in the Introduction and would support the biological plausibility of the study's hypothesis. The rationale for excluding CRP from the multivariate analysis of overall survival (OS) and disease-free survival (DFS) should also be clarified.

In Figure 2, the meaning of the term "Number at risk" is ambiguous. It should be specified whether this refers to the number of patients still under observation at each time point or to those who have survived up to that time. Additionally, the authors should address why CRP did not emerge as a prognostic factor alongside the DII.

Finally, the Conclusion section should be revised to more accurately reflect and interpret the findings presented in the study.

Author Response

  1. The Introduction section could benefit from a more comprehensive overview of previous related studies, potentially including other cancer types, to better contextualize the current work. Additionally, the authors should clearly articulate the unique contribution of their study to the existing literature.
    : Thanks to your thoughtful comment, we’ve added a few more references to studies on other cancer types in the Introduction section. (line 67-69)
    “The DII has been employed in various studies addressing diseases where inflammation plays a key role, including obesity, type 2 diabetes, cardiovascular disease, and various types of cancer.”
    Additionally, we have added the following paragraph to better highlight the unique contribution of our study. (line 74-77)
    “Given the current lack of evidence to guide dietary recommendations for CRC patients, particularly concerning dietary patterns that may enhance postoperative recovery and long-term prognosis, our study addresses this critical gap. Our findings aim to establish an evidence-based foundation for dietary guidance in the management of CRC patients.”

  2. In the Materials and Methods section, the description of the DII calculation method lacks sufficient detail and does not fully adhere to the journal’s methodological reporting guidelines. For enhanced clarity, it is recommended that an example of the food frequency questionnaire (FFQ) used in the study be included in the Supplementary Materials.
    : We appreciate the reviewer's suggestions for improvement. To provide a more detailed explanation of the DII calculation, we have revised certain parts of the text and added additional sentences. The changes are as follows. (line 121-131)
    The brief DII calculation process is as follows: (1) Individual intake data obtained from the FFQ arestandardized using Z-scores, calculated relative to a global reference database. The reference means and standard deviations for daily intake were developed by the original author using aggregated dietary data from 11 different countries. (2) Following conversion to percentile scores, Z scores are further transformed into centered percentile values, generating a symmetric distribution centered at zero and confined within the range of -1 to +1. (3) The centered percentile for each food parameter is multiplied by its corresponding inflammatory effect score, which was developed by the original author through an extensive literature review, to calculate the food parameter-specific DII score. (4) Summation of the DII scores for all individual food parameters yields the total DII score.
    Furthermore, thanks to your recommendation, we have included the FFQ used in this study as part of the Supplementary Materials.

  3. Furthermore, it would have been valuable to assess preoperative levels of pro- and anti-inflammatory cytokines, as these are referenced in the Introduction and would support the biological plausibility of the study's hypothesis. The rationale for excluding CRP from the multivariate analysis of overall survival (OS) and disease-free survival (DFS) should also be clarified.
    : We sincerely appreciate your insightful comment. Due to the retrospective design of our study, we were only able to use the preoperative CRP values available, which showed no significant difference between the two groups. However, since the DII is based on its relationship with six inflammatory biomarkers (including CRP), we believe that the association may not have been evident when looking at CRP alone. We agree that evaluating all six inflammatory biomarkers in a prospective study would be essential to further clarify the association between DII and inflammation.
    Furthermore, in multivariate analysis, we included only the variables with p-values less than 0.1 in the univariable analysis. This approach was taken to maintain statistical significance while avoiding the inclusion of an excessive number of variables in the multivariable model. As all variables were treated with equal consideration, we believe that the exclusion of CRP is justifiable.

  4. In Figure 2, the meaning of the term "Number at risk" is ambiguous. It should be specified whether this refers to the number of patients still under observation at each time point or to those who have survived up to that time. Additionally, the authors should address why CRP did not emerge as a prognostic factor alongside the DII.
    : We appreciate the reviewer's feedback. To reduce ambiguity, we have added an explanation for the Number at risk in Materials and Methods section. (line 163-165)
    “The number at risk represents the number of individuals who remain under follow-up at each time point and have not yet experienced the event of interest.”
    Meanwhile, as we mentioned in comment 3, since the DII is based on its relationship with six inflammatory biomarkers (including CRP), we believe that while the DII was confirmed as an independent prognostic factor, CRP alone may not have demonstrated the same prognostic significance.

  5. Finally, the Conclusion section should be revised to more accurately reflect and interpret the findings presented in the study.
    : We sincerely appreciate your helpful advice. We have presented the key findings of our study as follows. (line 350-352)
    “A high preoperative DII was associated with a higher incidence of postoperative complications and worse oncologic outcomes. Furthermore, it remained an independent prognostic factor for overall survival in the multivariate analysis.”

Reviewer 3 Report

Comments and Suggestions for Authors

This study aimed to investigate the impact of preoperative inflammatory diet, measured by the Dietary Inflammatory Index (DII), on clinical and oncologic outcomes following colorectal cancer surgery. The researchers included 126 patients who underwent colorectal cancer surgery between January 2018 and June 2020. The DII score was calculated using a 109-item semi-quantitative Food Frequency Questionnaire (FFQ), and patients were divided into high-DII and low-DII groups. The results showed that the high-DII group had an older age, a higher risk of postoperative complications within 30 days, and a lower 5-year overall survival rate. Multivariate survival analysis identified low-DII and M1 stage as independent prognostic factors for overall survival, and perineural invasion as an independent prognostic factor for disease-free survival. However, further revisions are still needed.

  • Should postoperative inflammatory diet also be included in the research?
  • The study used a 109-item FFQ to calculate the DII score, but only 26 food components (pre-set by the software) were actually used in the analysis. This selection of food components lacks reasonable representativeness.
  • Defects in FFQ Research: There is a contradiction regarding the FFQ in the study. In section 2.4, it states that if dietary changes were noted, participants were instructed to respond based on their dietary patterns from the year prior to the changes. However, in section 4. Discussion, it is proposed that for optimal results, patients with significant past dietary pattern changes should be excluded from the study, and annual follow-up FFQs should be conducted to confirm dietary pattern changes. This inconsistency needs to be resolved.
  • The relatively small sample size and insufficient FFQ analytical parameters limit the generalizability of the study's conclusions. A larger and more diverse sample is needed to draw a reliable conclusion applicable to a wider population.
  • The workload is insufficient. The study still needs to explore the biological molecular mechanisms by which inflammatory diet affects clinical and oncological outcomes.

Author Response

  1. Should postoperative inflammatory diet also be included in the research?
    : We sincerely appreciate your valuable insight, which has provided us with an important topic for further consideration. This study focused not only on long-term survival outcomes but also on short-term clinical outcomes; therefore, postoperative inflammatory diet was not considered in the analysis. However, investigating the association between postoperative inflammatory diet and various long-term outcomes would be a valuable direction for future research. We incorporated this consideration as a recommendation for future research in our study. (line 346-348)
    “Furthermore, as this study did not examine the role of postoperative dietary inflammation or dietary modifications, future research is needed to address these aspects.”

  2. The study used a 109-item FFQ to calculate the DII score, but only 26 food components (pre-set by the software) were actually used in the analysis. This selection of food components lacks reasonable representativeness.
    : We appreciate your careful attention to this critical aspect. Since the vast majority of patients treated at our institution are Korean adults, we determined that it would be most appropriate to employ an FFQ designed to reflect the typical dietary patterns of the Korean adult population. The 109-item FFQ we used includes food items commonly consumed by Korean adults, such as 'cooked rice,' 'bean paste soup,' 'bean curd,' and 'Korean cabbage kimchi.' When analyzed using the CAN-Pro 4.0 software, it provides raw intake amounts of various nutrients at a finer level, including carbohydrate, fat, protein, fiber, β-carotene, vitamin E, calcium, magnesium, chloride, cholesterol, butyric acid, palmitic acid, leucine, phenylalanine, etc. Among these, only 26 nutrients matched the food parameters included in the DII. No selection was made by the authors during this process.

  3. Defects in FFQ Research: There is a contradiction regarding the FFQ in the study. In section 2.4, it states that if dietary changes were noted, participants were instructed to respond based on their dietary patterns from the year prior to the changes. However, in section 4. Discussion, it is proposed that for optimal results, patients with significant past dietary pattern changes should be excluded from the study, and annual follow-up FFQs should be conducted to confirm dietary pattern changes. This inconsistency needs to be resolved.
    : We sincerely appreciate your insightful comment. As mentioned in Section 2.4, the FFQ is designed to assess food intake specifically over the past one year, and it can identify dietary changes that occurred within that one-year period. In the Discussion section, our intention was to emphasize that participants who had made dietary changes ‘more than one year prior’ should have been excluded from the study. We appreciate your comment, which helped us revise the expression to ensure greater clarity and avoid potential misunderstandings. (line 340-346)
    Finally, because the FFQ assesses dietary patterns only over the past year and can detect dietary changes only if they occurred within that specific timeframe, it cannot reflect the inflammatory potential of dietary patterns prior to that period or after FFQ administration. Therefore, for optimal results, patients who have undergone significant changes in their dietary patterns before the recall period need to be excluded from the study, and follow-up FFQs must be conducted annually to confirm whether the dietary pattern has undergone any changes.

  4. The relatively small sample size and insufficient FFQ analytical parameters limit the generalizability of the study's conclusions. A larger and more diverse sample is needed to draw a reliable conclusion applicable to a wider population.
    : We sincerely appreciate your insightful comment. During the initial study design phase, we performed a sample size calculation using the ‘ssizeCT.default’ function in R. Based on referenced values, we set pC(probability of failure in the control group over the maximum time period of the study) at 0.37 [1] and the relative risk (RR) at 0.49 [2], which indicated that a total study population of 232 participants would be required. Anticipating a response rate of approximately 50% for the food frequency questionnaire (FFQ), we aimed to recruit 400–500 patients in total.
    Throughout the study period, we recommended the FFQ to all eligible patients. However, given the extensive number of items (109 questions) and the estimated completion time of around 30 minutes, many patients declined to participate, resulting in a lower-than-expected response rate despite our efforts.
    To address the limitation of the insufficient sample size, we employed the Contal and O'Quigley method, a statistical technique that identifies the optimal cut-off value to maximize the survival difference between the two groups.
    Reference
    [1] Siegel RL, Giaquinto AN, Jemal A. Cancer statistics, 2024. CA Cancer J Clin. 2024 Jan-Feb;74(1):12-49. doi: 10.3322/caac.21820. Epub 2024 Jan 17. Erratum in: CA Cancer J Clin. 2024 Mar-Apr;74(2):203. doi: 10.3322/caac.21830. PMID: 38230766.
    [2] Zheng J, Tabung FK, Zhang J, Murphy EA, Shivappa N, Ockene JK, Caan B, Kroenke CH, Hébert JR, Steck SE. Post-cancer diagnosis dietary inflammatory potential is associated with survival among women diagnosed with colorectal cancer in the Women's Health Initiative. Eur J Nutr. 2020 Apr;59(3):965-977. doi: 10.1007/s00394-019-01956-z. Epub 2019 Apr 6. PMID: 30955051; PMCID: PMC6778721.

  5. The workload is insufficient. The study still needs to explore the biological molecular mechanisms by which inflammatory diet affects clinical and oncological outcomes.
    : We appreciate this important comment. We fully agree that further research is needed to elucidate the biological and molecular mechanisms underlying inflammatory diets. Future studies incorporating molecular analyses using cytokines or tumor samples, as well as investigations into the gut microbiome, appear to be warranted. We have included this as a recommendation for future research in our study. (line 336-340)
    Third, this study focused solely on the clinical and oncologic outcomes of CRC, without addressing the biological and molecular mechanisms underlying the impact of inflammatory diets. Future research should include molecular studies using tumor samples or animal models, as well as investigations at the microbiome level.

Round 2

Reviewer 2 Report

Comments and Suggestions for Authors

While I appreciate the authors' attempt to address my previous comment regarding the Introduction, I find the current revision insufficient. My original suggestion aimed to encourage the inclusion of a more comprehensive and concrete overview of prior studies involving the Dietary Inflammatory Index (DII), particularly in the context of cancer. The current addition “The DII has been employed in various studies addressing diseases where inflammation plays a key role, including obesity, type 2 diabetes, cardiovascular disease, and various types of cancer [21–25]” remains too general and lacks the critical synthesis expected in a scientific introduction.

For improved clarity and scientific rigor, I strongly encourage the authors to elaborate on the cited references [21–25], briefly summarizing the key findings, especially those related to cancer. This would not only demonstrate the relevance of the DII in oncology-related research but also better contextualize the novelty and rationale of the present study.

A dedicated paragraph synthesizing how DII has been previously linked to cancer risk, progression, or prognosis along with highlighting gaps or inconsistencies in the literature would substantially strengthen the Introduction.

I acknowledge the authors' effort to address my previous suggestion by including the food frequency questionnaire (FFQ) in the Supplementary Materials. However, I must point out that the version provided is in Korean, which significantly limits its utility for the international readership of this journal. Given that the vast majority of readers are unlikely to be fluent in Korean, the inclusion of the FFQ in its original language does not fulfill the intended purpose of enhancing clarity and transparency.

To ensure accessibility and reproducibility, I strongly recommend that the authors provide at least a translated version of the FFQ in English accompanied, if necessary, by a note indicating that it was originally administered in Korean. This would allow readers to better understand the dietary assessment methodology and facilitate future cross-study comparisons.

I appreciate the authors’ detailed response regarding the limitations associated with the retrospective design and the criteria applied for the multivariate analysis. However, I would recommend that the rationale for excluding CRP from the multivariate model despite its relevance to the study’s inflammatory framework be clearly stated in the Discussion section of the manuscript.

Specifically, it would be valuable to include a brief commentary on the lack of significant differences in CRP levels between the study groups, as this finding may be relevant to the interpretation of DII as a composite inflammatory marker. Incorporating this information into the main text would enhance transparency and provide a more complete understanding of why CRP did not emerge as a prognostic factor alongside the DII.

Author Response

  1. While I appreciate the authors' attempt to address my previous comment regarding the Introduction, I find the current revision insufficient. My original suggestion aimed to encourage the inclusion of a more comprehensive and concrete overview of prior studies involving the Dietary Inflammatory Index (DII), particularly in the context of cancer. The current addition “The DII has been employed in various studies addressing diseases where inflammation plays a key role, including obesity, type 2 diabetes, cardiovascular disease, and various types of cancer [21–25]” remains too general and lacks the critical synthesis expected in a scientific introduction. For improved clarity and scientific rigor, I strongly encourage the authors to elaborate on the cited references [21–25], briefly summarizing the key findings, especially those related to cancer. This would not only demonstrate the relevance of the DII in oncology-related research but also better contextualize the novelty and rationale of the present study. A dedicated paragraph synthesizing how DII has been previously linked to cancer risk, progression, or prognosis along with highlighting gaps or inconsistencies in the literature would substantially strengthen the Introduction.

    : We sincerely appreciate the reviewer's suggestions for improvement. As advised, we have added sentences summarizing the key findings of the referenced cancer-related studies. Furthermore, to better contextualize the novelty and rationale of the present study and to emphasize its significance, we have revised the final paragraph of the Introduction section as follows. (line 66-79)
    “In the field of oncology, it has been extensively studied as a potential risk factor for various malignancies, such as pancreatic, breast, and colorectal cancers [24-27].
    Given the pivotal role of prognostic factors in guiding clinical decisions and improving patient outcomes, continued research in this area is both necessary and timely. While prior studies have predominantly focused on clinical and pathological factors, there has been a progressive expansion of studies investigating the role of nutritional factors, such as sarcopenia, visceral obesity, and diet patterns, in prognostic evaluation. Among these, the DII has emerged as a potential prognostic indicator in CRC. However, comprehensive studies that integrate DII with the various spectrum of clinicopathologic variables remain limited [28-32]. Therefore, we aimed to determine the prognostic significance of DII by performing a multivariate analysis on prospectively collected clinicopathologic and nutritional data. The results of this study may provide a foundation for evidence-based dietary guidance in the management of patients with CRC.”

  1. I acknowledge the authors' effort to address my previous suggestion by including the food frequency questionnaire (FFQ) in the Supplementary Materials. However, I must point out that the version provided is in Korean, which significantly limits its utility for the international readership of this journal. Given that the vast majority of readers are unlikely to be fluent in Korean, the inclusion of the FFQ in its original language does not fulfill the intended purpose of enhancing clarity and transparency. To ensure accessibility and reproducibility, I strongly recommend that the authors provide at least a translated version of the FFQ in English accompanied, if necessary, by a note indicating that it was originally administered in Korean. This would allow readers to better understand the dietary assessment methodology and facilitate future cross-study comparisons.

    : Thank you for your valuable comment. We agree with your suggestion and have newly included the English-translated version of the FFQ as a Supplementary Material.

  1. I appreciate the authors’ detailed response regarding the limitations associated with the retrospective design and the criteria applied for the multivariate analysis. However, I would recommend that the rationale for excluding CRP from the multivariate model despite its relevance to the study’s inflammatory framework be clearly stated in the Discussion section of the manuscript. Specifically, it would be valuable to include a brief commentary on the lack of significant differences in CRP levels between the study groups, as this finding may be relevant to the interpretation of DII as a composite inflammatory marker. Incorporating this information into the main text would enhance transparency and provide a more complete understanding of why CRP did not emerge as a prognostic factor alongside the DII.

    : Thanks to your thoughtful comment. We have incorporated this content into the Discussion section as follows. (line 297-305)
    “Preoperative CRP levels are generally known to be associated with survival in patients with CRC [41,42]. However, in this study, preoperative CRP levels did not show a statistically significant difference between the high-DII and low-DII groups. Additionally, in the univariate survival analysis, CRP did not demonstrate prognostic significance, similar to the DII. However, as the DII is based on its relationship with six inflammatory cytokines, the interpretation of CRP alone may be limited. Previous studies have also reported inconsistent findings, with some demonstrating a significant correlation between CRP and the DII, while others have found no meaningful association [43,44]. It is necessary to conduct further studies that incorporate all six inflammatory cytokines.”

Reviewer 3 Report

Comments and Suggestions for Authors

The author has answered my question very well and has also conducted careful consideration and detailed revision on the question raised.

Author Response

  1. The author has answered my question very well and has also conducted careful consideration and detailed revision on the question raised.

    : We are deeply grateful for the reviewer’s detailed and thoughtful comments. Thanks to the reviewer’s valuable feedback, we were able to address the shortcomings and complete a more refined version of our manuscript.